# Avian antibodies (IgY) targeting spike glycoprotein of severe acute respiratory syndrome coronavirus 2 (SARS-CoV-2) inhibit receptor binding and viral replication

Chad Artman[1☯], Kyle D. Brumfield[2,3☯], Sahil Khanna[4], Julius Goepp[1]*

1 Scaled Microbiomics, LLC, Hagerstown, MD, United States of America, 2 Maryland Pathogen Research Institute, University of Maryland, College Park, MD, United States of America, 3 University of Maryland Institute for Advanced Computer Studies, University of Maryland, College Park, MD, United States of America, 4 Division of Gastroenterology and Hepatology, Mayo Clinic, Rochester, MN, United States of America

☯ These authors contributed equally to this work.
* jgoepp@scaledmicrobiomics.com

**Data Availability Statement:** All relevant data are within the paper and its Supporting Information files.

## Abstract

### Background

The global pandemic of Coronavirus infectious disease 2019 (COVID-19), caused by SARS-CoV-2, has plunged the world into both social and economic disarray, with vaccines still emerging and a continued paucity of personal protective equipment; the pandemic has also highlighted the potential for rapid emergence of aggressive respiratory pathogens and the need for preparedness. Avian immunoglobulins (IgY) have been previously shown in animal models to protect against new infection and mitigate established infection when applied intranasally. We carried out a proof-of-concept study to address the feasibility of using such antibodies as mucosally-applied prophylaxis against SARS-CoV-2.

### Methods

Hens were immunized with recombinant S1 spike glycoprotein of the virus, and the resulting IgY was evaluated for binding specificity, inhibition of glycoprotein binding to angiotensin converting enzyme-2 (ACE2) protein (the requisite binding site for the virus), and inhibition of viral replication in Vero cell culture.

### Results

Titers of anti-S1 glycoprotein IgY were evident in yolks at 14 days post-immunization, peaking at 21 days, and at peak concentrations of 16.8 mg/ml. IgY showed strong and significant inhibition of S1/ACE2 binding interactions, and significantly inhibited viral replication at a concentration of 16.8 mg/ml. Four weeks' collection from eggs of two hens produced a total of 1.55 grams of IgY.

**Funding:** This study was funded by Scaled Microbiomics, LLC. The funder provided support in the form of salaries for authors CA and JG, but did not have any additional role in the study design, data collection and analysis, decision to publish, or preparation of the manuscript. The specific roles of these authors are articulated in the "author contributions" section.

**Competing interests:** Scaled Microbiomics, LLC (SMB) provided sole funding for this study. Authors CA and JG are employees of SMB; JG is also Founder and CEO of SMB. This does not alter our adherence to PLOS ONE policies on sharing data and materials. No competing interests exist in relation to any authors' affiliation with Scaled Microbiomics.

## Conclusions

In this proof-of-concept study we showed that avian immunoglobulins (IgY) raised against a key virulence factor of the SARS-CoV-2 virus successfully inhibited the critical initial adhesion of viral spike glycoproteins to human ACE2 protein receptors and inhibited viral replication *in vitro*, in a short period using only two laying hens. We conclude that production of large amounts of IgY inhibiting viral binding and replication of SARS-CoV-2 is feasible, and that incorporation of this or similar material into an intranasal spray and/or other mucosal protecting products may be effective at reducing infection and spread of COVID-19.

## Introduction

The global pandemic of Coronavirus infectious disease 2019 (COVID-19), caused by several variants of the SARS-CoV-2 virus, has plunged the world into both social and economic disarray. The recent emergence of several efficacious vaccines will mitigate this impact, but full population-level coverage remains in the future. Emergence of new strains may further complicate attainment of widespread immunity to this virus and its variants. Finally, the absence of effective non-vaccine countermeasures highlights concerns about future outbreaks of emerging viral respiratory illnesses. Under these circumstances, an effective, rapidly-deployable, non-vaccine prophylactic approach, especially one that could be stockpiled ahead of new outbreaks, may add new tools to the public-health preparedness armamentarium.

Passive immunization using pathogen-specific antibodies is a potential means of preventing disease by various infectious agents in both humans and animals [1–3]. A critical necessity in this approach, however, is the production and availability of large quantities of inexpensive antibodies. Therefore, polyclonal antibodies may be advantageous over monoclonal antibodies, which, while highly epitope-specific, may not offer full clinical protection and are costly to produce in bulk [4].

Immunoglobulin Y (IgY), the primary circulating antibody in the serum of avians, reptiles, and amphibians, is transferred to yolks of developing eggs and provides passive immunity to the newly-hatched young [4, 5]. Readily harvested at high yields from immunized hens' eggs, IgY has been shown to be safe and effective at both prophylaxis and treatment of bacterial and viral infections in laboratory and agricultural animals [1, 4]. A small number of human studies have shown IgY to be safe and effective in human therapeutic applications [2, 3, 6–8].

With regard to respiratory viral infections, intranasal application of IgY targeting whole, inactivated influenza virus has been shown to prevent murine influenza viral infection when applied prior to viral challenge, and to mitigate influenza disease when applied within a limited period following challenge [9, 10]. Human studies have demonstrated safety and efficacy of prophylactic intra-oral IgY against *Pseudomonas aeruginosa* in cystic fibrosis patients, and nonspecific intranasal polyclonal IgA in upper respiratory disease [11–14].

Given high transmissibility, lack of either a vaccine or an effective treatment, high mortality with the current COVID-19 crisis, and the threat to economic wellbeing posed by extended closures of businesses and public services, we conducted a proof-of-concept study of IgY targeting SARS-CoV-2 as a potential mucosal prophylaxis against infection. Results showed inhibition of binding to Angiotensin-2 (ACE2) receptor protein and reduced viral replication in laboratory studies, and feasibility studies of mucosal prophylactic formulation *in vivo* are in progress.

## Materials and methods

### Immunization of laying hens

Recombinant SARS-CoV-2 spike glycoprotein (S1) with sheep Fc-tag (HEK293; The Native Antigen Company, Oxford, United Kingdom) was reconstituted in phosphate-buffered saline (PBS) to 10% w/v and mixed under high-shear conditions with the poultry adjuvant Montanide ISA 70 VG (30% antigen in PBS, 70% adjuvant, v/v), following manufacturer's specifications (Seppic Inc, Fairfield, NJ USA). The resulting emulsion, containing 100 μg spike glycoprotein/ml, was filter-sterilized using a 0.22 μm pore size polycarbonate filter membrane (VWR International, Radnor, PA, USA). Commercial White Rock and Rhode Island cross-bred, sexlink hens (Pinola Hatchery, Shippensburg, PA, USA) were maintained segregated in three pairs and acclimated for two weeks prior to immunization in a protected coop at ambient temperatures on a 12-hour light/dark cycle on *ad libitum* water and a commercial diet (Martin's Layer Mash 16%, Martin's Elevator, Inc., Hagerstown, MD, USA).

On immunization day 1, 0.5 ml (50 mcg antigen) of the vaccine/adjuvant preparation was injected intramuscularly into each breast of each of one hen pair (test pair). For control, two hens were injected with a PBS/adjuvant (30%/70% v/v) mixture and an additional pair of hens remained unimmunized. On immunization day 14, booster immunizations were performed identically to immunization day 1 for each hen pair. Protocols for hen maintenance, immunization, and phlebotomy were approved by the Scaled Microbiomics, LLC Animal Use and Care Committee (approval number 2020–001).

### Serum IgY antibody determination

To detect circulating anti-S1 Spike antibody levels, 4 mL whole blood was drawn by brachial venipuncture from individual hens at 3 and 4 weeks after the first immunization, using 23G x ¾" SecureTouch safety scalp vein butterfly sets (Exel International, Quebec, Canada). To allow for clot formation, blood samples were incubated at room temperature (23°C– 25°C) for 30 minutes and stored at 4°C overnight (16 h). Blood serum was separated from cells by centrifugation at 5,000g for 5 minutes, and IgY titer (from both immunized and unimmunized hens) was determined by enzyme-linked immunosorbent assay (ELISA) against the S1 antigen as described below. Samples were only obtained at Weeks 3 and 4; serum collection was discontinued after Week 4 when yolk IgY levels were detected.

### IgY extraction and concentration

For IgY egg yolk extraction, two eggs were collected weekly from each pair of hens beginning the first day prior to first immunization. IgY was extracted from yolks using polyethylene glycol (PEG) as described elsewhere [15], with the following modifications. Briefly, yolks were pooled, and lipid content was removed by centrifugation (13,000 x g for 20 min at 4°C) using PEG 6000 (Alfa Aesar, Haverhill, MA, USA) at 12% w/v. The resulting precipitate was resuspended in PBS and dialyzed against dilute sodium chloride (0.1% w/v) overnight (16 h) and then against PBS for three hours.

Total protein concentration of egg yolk and serum was determined by bicinchoninic acid method kit, following manufacturer's specifications (Thermo Fisher Scientific, Rockford, IL, USA), using S1-coated 96-well plates. Absorbance values were read at 490 nm, and standard curve showed linear behavior over seven serial 1:2 dilutions ($R^2$ = 0.99; **S1 Fig**) using bovine gamma-globulin (Bio-Rad, Hercules, CA, USA) as the protein standard set. Resulting IgY was stored at -20°C until further analysis (< 2 weeks).

## Sodium dodecyl sulfate-polyacrylamide gel electrophoresis (SDS-PAGE)

To determine purity of both yolk- and serum-derived IgY, SDS-PAGE was conducted under reducing conditions using 12% polyacrylamide gel (NuSep Inc., Germantown, MD, USA) with a Novex Mini-Cell (Invitrogen, Carlsbad, CA, USA). Briefly, the sample was mixed with sample buffer and incubated for 5 min at 100˚C. A total of 10 μl of sample was loaded into each well. Precision Plus Dual Color Pre-stained protein standard (10-250kDa) (Bio-Rad, Hercules, CA, USA) was used as a molecular weight marker. The protein bands were visualized with Protein Fixative (Ward's Science, Rochester, NY, USA). The gel was imaged using a standard camera; the original gel image can be found in the **S1 Raw image**. Specificity of IgY binding to purified S1 glycoprotein was demonstrated by dot-blot (**S2 Fig**).

## Enzyme-linked immunosorbent assay

IgY titer against SARS-CoV-2 S1 glycoprotein subunit was measured by an indirect noncompetitive ELISA as reported previously with slight modifications [4, 16, 17]. Briefly, a 96-well microtiter plate was coated with S1 antigen at 4 ng/μl protein in carbonate coating solution (BioLegend, San Diego, California) using 100 μl/well. Plates were blocked using skim milk in PBS (5% w/v) overnight (16 h) at 4˚C. Serial dilutions (1:100, 1:1,000, 1:10,000) of anti-S1 IgY, obtained from both yolk and serum, were incubated in blocked plates for one hour at room temperature (23˚C-25˚C). Bound anti-S1 IgY was detected with 100 μl per well horseradish peroxidase-conjugated goat anti-chicken IgY (1:2,500) (ImmunoReagents, Inc., Raleigh, NC, USA). After incubation for 1 hour at room temperature (23˚C-25˚C), the plate was washed five times using commercial ELISA wash buffer (Thermo Fisher, Rockford, IL, USA). Next, 100 μl per well 3,3'-5,5'-tetramethylbenzidine (VWR International, Radnor, PA, USA) one-component substrate was added and incubated for 15 min at room temperature (23˚C-25˚C). The color development was stopped with 2N sulfuric acid (VWR International, Radnor, PA, USA) (100 μl per well), and the optical density (OD) was measured on a microtiter plate reader (Molecular Devices, Sunnyvale, California, USA) at 450 nm. Experiment reproducibility and assay calibration was established by including triplicate wells coated with S1 antigen (background from antigen noise) which received a blank IgY treatment (PBS with 5% w/v skim milk; No IgY), triplicate non-coated wells (background from IgY noise) which received a blank antigen treatment (carbonate coating buffer; No S1 antigen), and two negative unimmunized IgY experimental controls obtained from eggs (S1-vaccinated hen pre-immunization IgY and unimmunized control hen IgY) in each plate. SARS-CoV-2 S1-specific IgY titer was defined as the maximum dilution multiple of the sample with an OD value that was $\geq 2.1$ times that of both negative controls.

## Binding inhibition of S1 to ACE2

Binding interactions of S1 to ACE2 protein were analyzed using the SARS-CoV-2 Inhibitor Screening Kit (Acro Biosystems, Newark, DE), according to the manufacturer's instructions [18]. This kit provides the receptor binding domain (RBD) peptide of the S1 glycoprotein spike, and a standardized inhibitor of S1/ACE2 binding, which provides a positive control. Method Verification was carried out per the included kit instructions (**S1 File**) [18].

Additionally, to determine inhibitory capacity of anti-SARS-CoV-2 S1 IgY against the entire S1 glycoprotein, the assay was repeated with the following modifications. Briefly, recombinant SARS-CoV-2 spike glycoprotein (S1) with sheep Fc-tag (HEK293; The Native Antigen Company, Oxford, United Kingdom) was used in place of the S1 receptor binding domain protein provided by the kit.

## Plaque reduction assay

To evaluate the ability of S1-specific IgY to prevent viral replication in vitro, a plaque reduction assay was carried out. Briefly, anti-S1 and control (unimmunized) IgY were incubated for 1 hour at 37°C with SARS-Cov2 virus (BEI Resources, Manassas, VA, USA) at a 1:1 ratio (50 µl of undiluted antibody with 50 µl of virus with a titer of 3 x $10^5$ pfu/ml). Vero cells (ATCC) were seeded at a density of 3 x $10^5$ cells/ml and incubated overnight at 37°C until confluent. Antibody-viral mixtures were serially diluted (undiluted, 1:10, 1:100, 1:1000) in Dulbecco's modified Eagle's medium and used to infect Vero cells. Infected plates were incubated at 37°C for 1 hour, followed by addition of an agar overlay (2X Eagle's Minimum Essential medium and 0.06% agarose at a 1:1 ratio) into each well. Infected cells with no IgY treatment and cells treated with antibody from unimmunized hens' eggs were used as controls. Plates were incubated at 37°C in a 5% $CO_2$ cell culture incubator for 48 hours. Cells were then fixed with formaldehyde (10% v/v), added to plates with agar plugs, and incubated at room temperature (23°C-25°C) for 45 min. Plates were rinsed with deionized water, and agar plugs removed. Wells were stained using crystal violet (1% v/v) and washed with ethanol (20% v/v) twice. Plaques were counted to determine viral titer as plaque forming units (pfu/ml); raw data are available in S2 File.

# Results

## Isolation and purification of IgY

Avian IgY was successfully isolated from yolks, yielding on Day 28 following first immunization, up to 16.8l mg/ml (min. 10 mg/ml, avg. 12.4 mg/ml), and from serum, yielding, on Day 21 following first immunization, up to 23.1mg/ml (min. 20.9 mg/ml, avg. 22.1mg/ml). SDS-PAGE showed characteristic IgY banding patterns, with molecular weights of 68 kDa and 25 kDa, representing the IgY heavy and light chains, respectively (Fig 1). By the end of the 6-week study period, a total of 125 ml of IgY in PBS solution had been produced, yielding a total of 1.55 g IgY from two immunized hens.

## Titer of anti-S1 glycoprotein IgY in egg yolks

Titers of IgY against the S1 glycoprotein rose following two immunizations to a peak value of 1:10,000 by Week 3, falling to 1:1000 by Week 6 (Fig 2), similar to our observations in work with other viral and bacterial antigens (S3 Fig). Serum anti-S1 glycoprotein antibody levels were detected at weeks 3 and 4 at titers of 1:10,000, after which serum collection was discontinued.

## Inhibition of S1 spike binding to ACE2 protein by S1-specific IgY

Using the kit-provided S1 spike glycoprotein RBD peptide as substrate against the target ACE2 protein, binding was inhibited significantly in a concentration-dependent manner at IgY dilutions through 1:10 (undiluted, p = 1.2 x $10^{-5}$; 1:10, p = 1.3 x $10^{-4}$; Fig 3). The undiluted material used in this and subsequent analyses contained 16.8 mg IgY/ml.

When the binding inhibition assay was employed using the full S1 spike glycoprotein, binding of the spike glycoprotein to the target ACE2 protein was significantly inhibited in a concentration-dependent manner at all IgY dilutions through 1:100 (undiluted, p = 4.2 x $10^{-6}$; 1:10, p = 1.5 x $10^{-6}$; 1:100, p = 8.6 x $10^{-4}$; Fig 4).

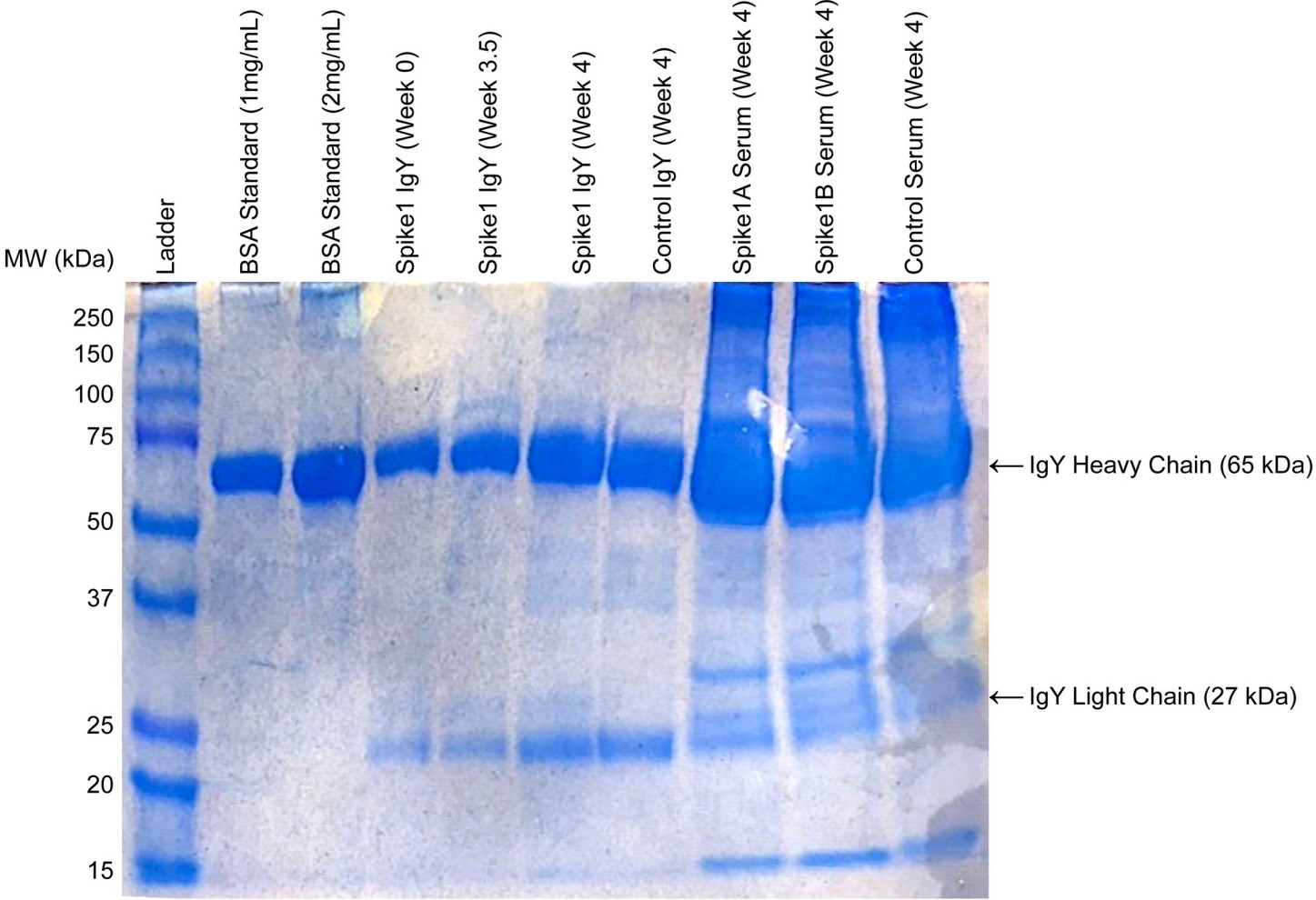

**Fig 1. SDS-PAGE analysis of SARS-CoV-2 S1 spike-specific IgY carried out under reducing conditions.** Lanes 4–7 show IgY extracted from yolks at indicated time intervals; lanes 8–10 show IgY in serum at Week 4; "A" and "B" indicate serum from each hen. IgY heavy and light chains show characteristic bands at 65 and 27 kDa, respectively. "Control" indicates material from unimmunized hens.

### Inhibition of SARS-CoV-2 virus replication by IgY

To further evaluate the antiviral activity of the S1-specific IgY, a plaque reduction assay was performed. The undiluted IgY at a protein concentration of 16.8 mg/ml produced a significant 38.3% reduction in plaque-forming units compared with control IgY (**Fig 5**).

### Discussion

SARS-CoV-2, the causative viral agent of COVID-19, is transmitted person-to-person primarily through close contact and respiratory droplets [20]. After entry to the human respiratory tract, the virus must bind to ACE2 protein receptors on respiratory epithelial cells [21]. These features of viral infection offer an opportunity for prevention of active COVID-19 by intercepting incoming viral particles and preventing adhesion, which is required for cellular invasion. While masks and social distancing play important roles in slowing individual infection and person-to-person spread, protection is incomplete, and these methods have substantial social and economic impact. An alternate, or additional, means of protection might be offered by prophylactic treatment of mucosal surfaces.

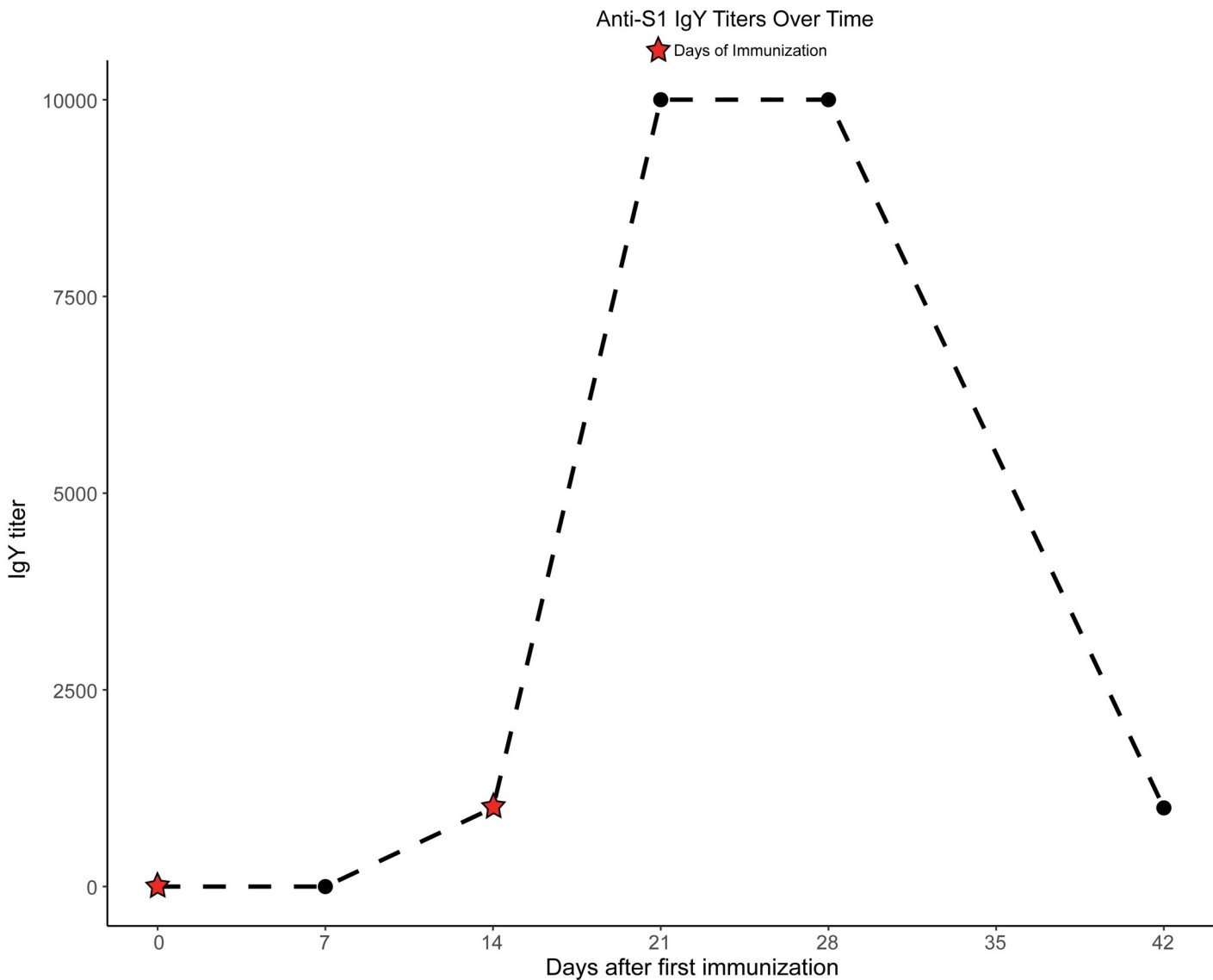

**Fig 2. Titers of anti-SARS-CoV-2 S1 spike IgY in egg yolks over time.** Eggs were collected from immunized hens weekly over 6 weeks following the first immunization and analyzed by ELISA against the S1 antigen. Stars indicate dates of immunization injections.

In this proof-of-concept study we demonstrated that IgY targeting the functionally-important S1 spike glycoprotein of SARS-CoV-2 can be produced rapidly after only two injections of laying hens with the intact glycoprotein as antigen, yielding titers of IgY antibody of 1:10,000 by 3 weeks following the first immunization (**Fig 2**). Titers fell by 6 weeks to levels consistent with those in prior studies of influenza antigens in which up to 5 immunizations were performed [4]. The resulting IgY was strongly inhibitory of the essential S1 spike/ACE2 protein binding interaction required for initiation of infection (**Figs 3** and **4**), and demonstrated significant viral neutralization in a plaque reduction assay carried out with active SARS-CoV-2 viral particles (**Fig 5**).

In the four-week egg collection period of this study, we collected a total of 1,550 mg polyclonal IgY produced by two immunized hens, using a simple physicochemical extraction PEG extraction method. Flocks of 100,000 hens are commonly used in commercial egg production

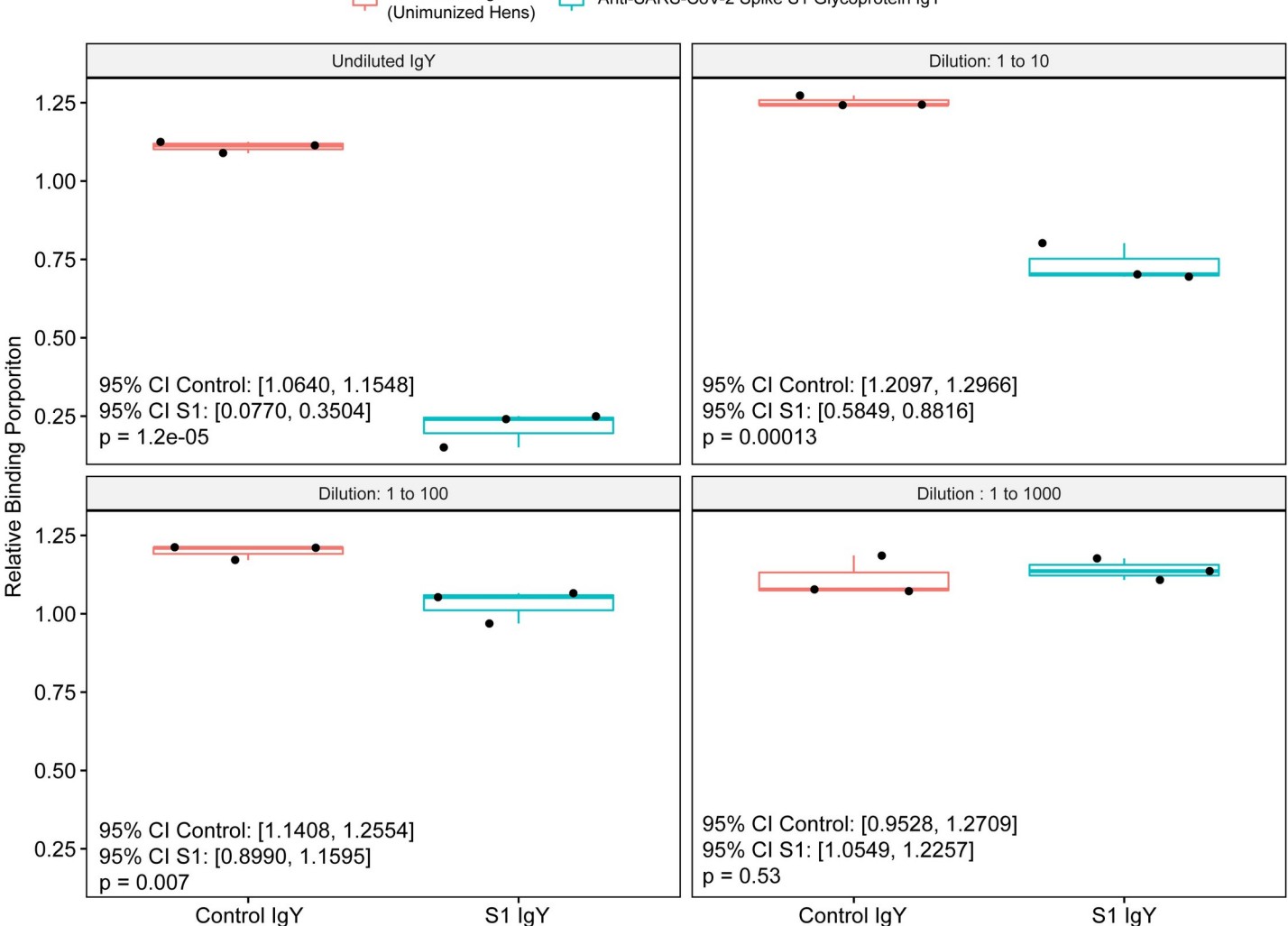

**Fig 3. Box and whisker plots of SARS-CoV-2 spike glycoprotein (S1) Receptor Binding Domain (RBD) percent binding to angiotensin converting enzyme-2 protein in a cell-free system.** Relative binding proportion (Y-axis) was determined by the kit-provided standard binding inhibitor. Boxes represent interquartile range (IQR) with median shown as center bar of each sample group. Whiskers represent 1.5 times the IQR. P-value, by two-sample t-test method, and 95% confidence interval (CI) was calculated using R software package EnvStats (v.2.3.1). [19] Undiluted (16.8 mg/ml), and logarithmic dilutions (1.68, 0.168, and 0.0168 mg/ml) are shown. Coral, control IgY, teal, anti-SARS-CoV-2 spike glycoprotein IgY.

facilities, suggesting that in excess of 1,000 kg of this material could be produced annually by one such flock in large-scale production.

IgY has seen use in studies of respiratory infections, both viral and bacterial. Orally administered IgY showed high safety and efficacy in preventing colonization with *Pseudomonas aeruginosa* in human cystic fibrosis patients [2, 3]. IgY raised in ostrich eggs against pandemic strains of influenza (A/H1N1) was shown to inhibit hemagglutination and viral replication of the cognate strain in vitro [22].

Studies by Wallach, Wen, and others have demonstrated that intranasally-applied aqueous solutions of IgY targeting influenza viruses (A and B) is completely protective against murine infection when applied up to 6 hours prior to lethal viral exposure, and to prevent mortality

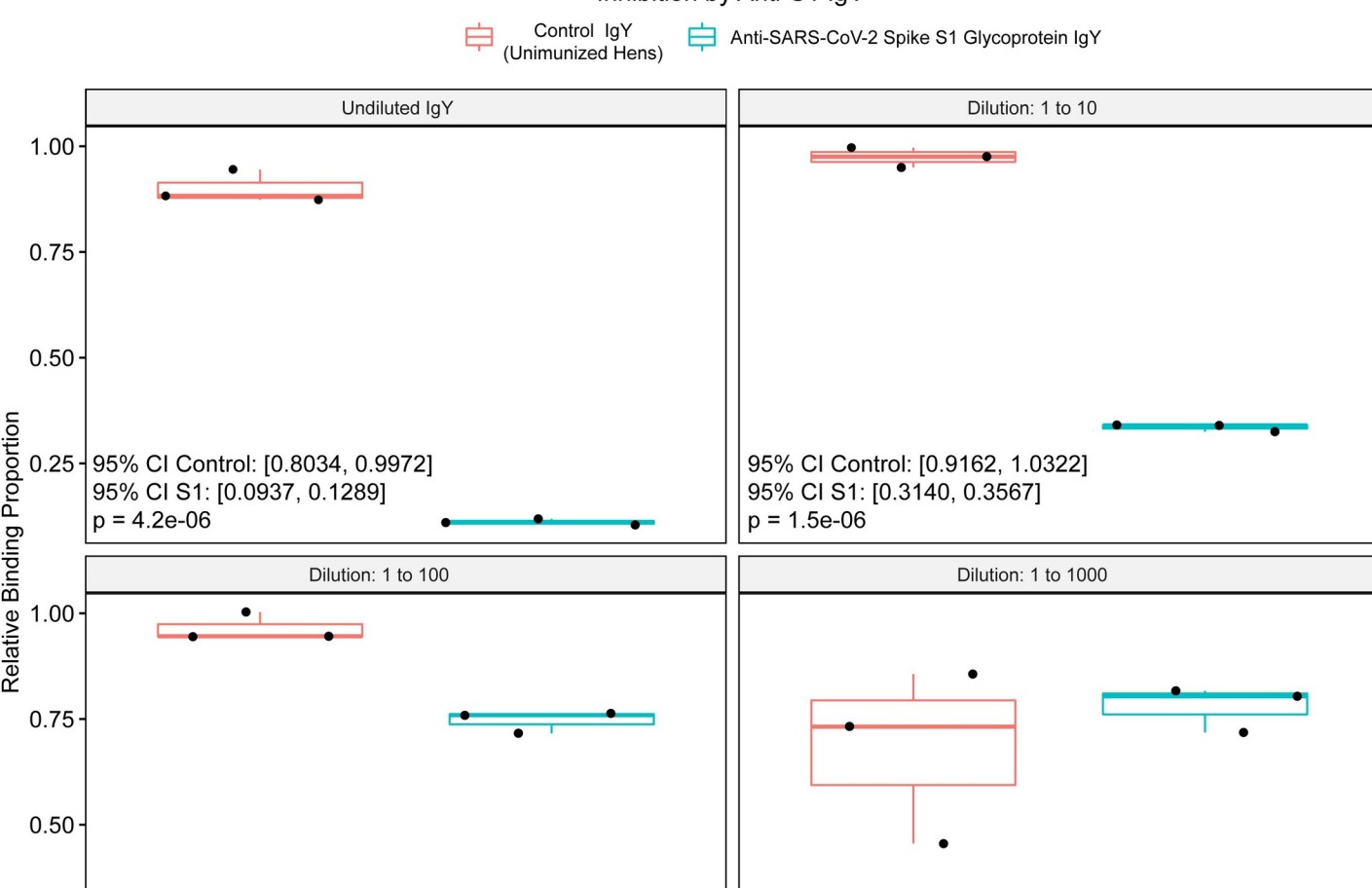

**Fig 4. Box and whisker plots of SARS-CoV-2 spike glycoprotein (S1) percent binding to angiotensin converting enzyme-2 protein in a cell-free system.** Relative binding proportion (Y-axis) was determined by the kit-provided standard binding inhibitor. Boxes represent interquartile range (IQR) with median shown as center bar of each sample group. Whiskers represent 1.5 times the IQR. P-value, by two-sample t-test method, and 95% confidence interval (CI) was calculated using R software package EnvStats (v.2.3.1) [19]. Undiluted (16.8 mg/ml), and logarithmic dilutions (1.68, 0.168, and 0.0168 mg/ml) are shown. Coral, control IgY, teal, anti-SARS-CoV-2 spike glycoprotein IgY.

when administered up to 16 hours post-exposure to a lethal dose of multiple viral strains [9, 10, 23]. Pathology examination of lung tissue from animals treated post-exposure showed significant reduction in lung pathology and viral replication compared with untreated mice [4, 23].

These findings, taken together with the work on IgY against SARS-CoV-2 presented here, suggest that an intranasal formulation of IgY targeting the SARS-CoV-2 virus may demonstrate similar prophylactic, and possibly therapeutic, results.

Given the large abundance of target-specific IgY in yolks of immunized hens' eggs, the speed at which large quantities can be produced, the history of safety and efficacy of intranasal IgY in prophylaxis and treatment of serious respiratory infections, the present lack of a vaccine

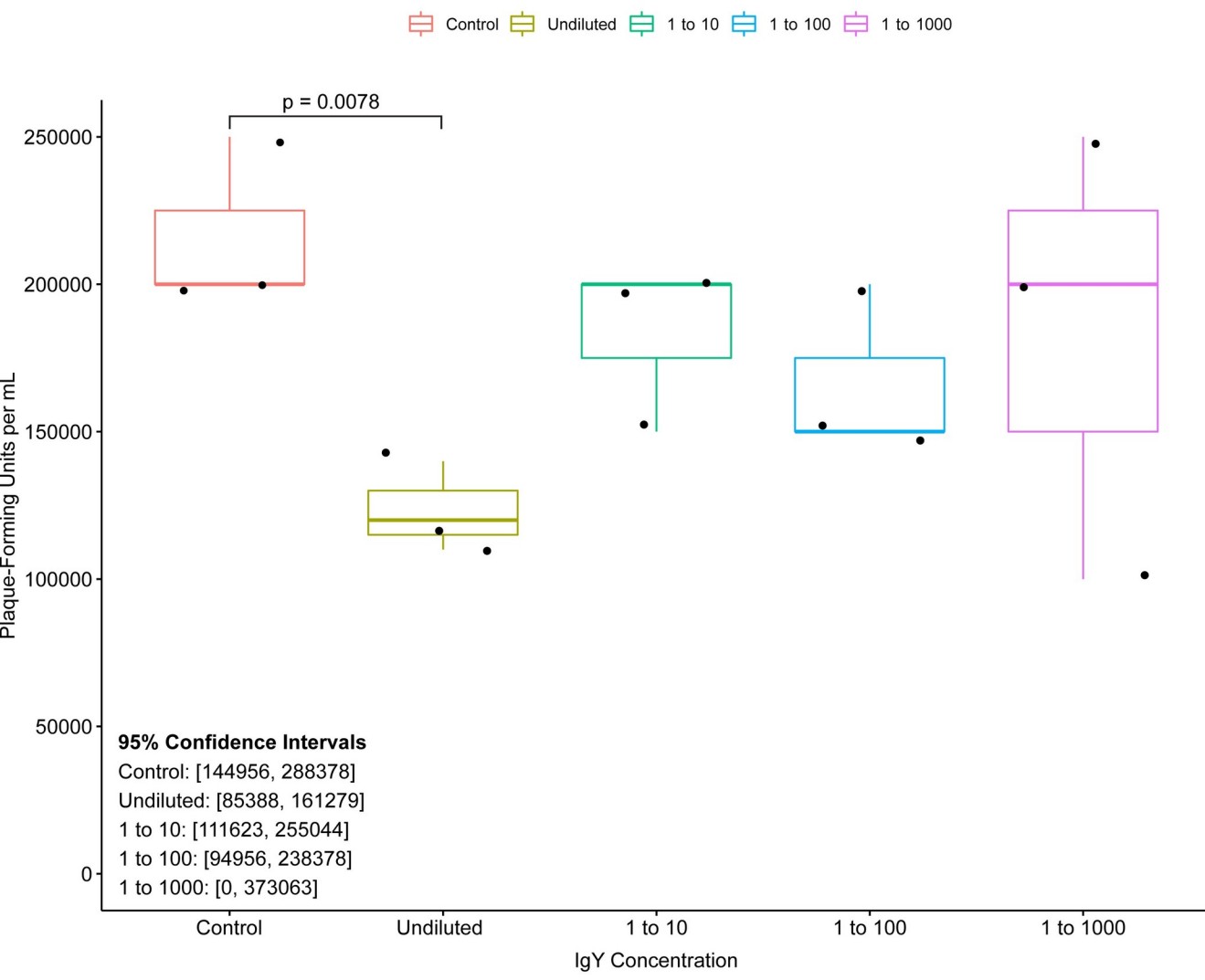

**Fig 5. Box and whisker plots of SARS-CoV-2 viral replication patterns *in vitro*.** Boxes represent interquartile range (IQR) with median shown as center bar of each sample group. Whiskers represent 1.5 times the IQR. P-value, by two-sample t-test method, and 95% confidence interval (CI) was calculated using R software package EnvStats (v.2.3.1) (Millard, 2013). Undiluted (16.8 mg/ml), and logarithmic dilutions (1.68, 0.168, and 0.0168 mg/ml) are shown. P-values less than 0.05 are shown. Coral, control IgY; gold, undiluted S1 IgY; green, S1 IgY diluted at 1:10; blue, S1 IgY diluted at 1:100; purple, S1 IgY diluted at 1:1000.

against SARS-CoV-2, and the imperative to prevent individual infections and person-to-person spread, we believe that further study of anti-SARS-CoV-2 IgY is warranted. Future studies should examine generation of IgY against RBD epitopes of the S1 glycoprotein, to reduce production costs and increase target specificity.

Animal studies are underway to determine *in vivo* efficacy; human studies to establish safety and tolerability of a final formulation will also be required, and regulatory pathways should also be explored. SARS-CoV-2 is not exclusively transmitted by nasal inhalation; the active IgY antibodies studied here are readily incorporated into oral lozenges or throat sprays, and can be formulated into sterile ophthalmic solutions as needed.

This proof-of-concept study has certain limitations. The number of hens used to generate anti-SARS-CoV-2 spike IgY was small, which may limit generalizability. Our data, however, show only minimal differences in IgY levels between hens (data not shown). We used ordinary agricultural laying hens, rather than specific pathogen-free birds, leaving open the small possibility of inadvertent inclusions of potential pathogens that could be transmitted to human subjects; this can be addressed in future studies seeking evidence of pathogens in the IgY material collected. This study was designed as a brief proof-of-concept, so that we collected eggs for a limited period, and performed only two immunizations, limiting our ability to accurately predict commercial production levels. In our hands, a more typical immunization schedule of up to five injections at two-week intervals produces sustained high-level titers.

## Conclusions

In this proof-of-concept study we showed that avian immunoglobulins (IgY) raised against a key virulence factor of the SARS-CoV-2 virus successfully inhibited the critical initial adhesion of viral spike glycoproteins to human ACE2 protein receptors and inhibited viral replication *in vitro*, in a short period using only two laying hens. We conclude that production of large amounts of IgY inhibiting viral binding and replication of SARS-CoV-2 is feasible, and that incorporation of this or similar material into an intranasal spray and/or other mucosal protecting products may be effective at reducing infection and spread of COVID-19.

## Supporting information

**S1 Fig. Standard curve for bicinchoninic acid (BCA).** Total protein concentration was determined by BCA method targeting SARS-CoV-2 S1. Curve shows linearity over eight serial dilutions ($R^2 = 0.99$).
(TIF)

**S2 Fig. Dot-blot image showing IgY binding specificity against purified S1 glycoprotein.** Hens were immunized using norovirus virus-like particles (NVLP), and IgY was prepared as mentioned in the "Materials and methods" section. IgY solutions (NVLP, S1, and unimmunized) were spotted onto a nitrocellulose membrane (Azure Biosystems, Dublin, CA, USA) and allowed to dry at room temperature (RT) for 30 minutes. Each membrane was blocked in tris-buffered saline and Tween 20 (TBST) supplemented with 5% (v/v) skim milk for 1 hr at RT. Membranes were incubated with the appropriate IgY dilutions (1:1,000 anti-S1; 1:2,500 anti-VLP, and Unimmunized IgY) in blocking buffer at RT. After 1 hr, the primary antibody dilutions were aspirated, and each blot was washed three times (5 min) with TBST. Blots were then incubated for 1 hr with a 1:1,000 dilution of Goat anti-chicken HRP-conjugated IgG (ImmunoReagents, Inc., Raleigh, NC, USA) in blocking buffer at RT. Secondary antibody solution was then aspirated, and blots were washed three times (10 min followed by two 5 min washes) with TBST and washed a final time (5 min) with TBS. Color change was observed using TMB chemical substrate (VWR International, Radnor, PA, USA), per manufacturer's directions. The reaction was quenched using TBS, and images were captured using a standard camera.
(TIF)

**S3 Fig. IgY production over time.** ELISA titers ($\log_2$) of IgY targeting norovirus virus-like particles and bacterial multiepitope fusion antigen over 24 weeks are shown as line plots. Viral and bacterial immunizations were carried out intramuscularly, as mentioned in the "Materials and methods" section, at weeks 0, 2, and 4.
(TIF)

**S1 File. SARS-CoV-2 Inhibitor Screening Kit method verification.** SARS-CoV-2 inhibitor screen kit method verification was performed following the manufacturer's specifications [18].
(DOCX)

**S2 File. Raw data file for plaque reduction assay.**
(XLSX)

**S1 Raw image. SDS-PAGE analysis of SARS-CoV-2 S1 spike-specific IgY carried out under reducing conditions–raw image.**
(PDF)

## Acknowledgments

The authors recognize the efforts of our Laboratory Director, Mohamed Ait Ichou, PhD, Laboratory Technicians Nnebuefe Idegwu and Drupad Patel, and David Myers, who provided animal husbandry of laying hens and who carried out yolk harvests.

## Author Contributions

**Conceptualization:** Kyle D. Brumfield, Julius Goepp.

**Data curation:** Chad Artman, Kyle D. Brumfield.

**Formal analysis:** Chad Artman, Kyle D. Brumfield.

**Funding acquisition:** Julius Goepp.

**Investigation:** Chad Artman, Julius Goepp.

**Methodology:** Chad Artman, Kyle D. Brumfield, Julius Goepp.

**Project administration:** Julius Goepp.

**Resources:** Julius Goepp.

**Software:** Kyle D. Brumfield.

**Supervision:** Julius Goepp.

**Validation:** Kyle D. Brumfield.

**Visualization:** Kyle D. Brumfield.

**Writing – original draft:** Julius Goepp.

**Writing – review & editing:** Chad Artman, Kyle D. Brumfield, Sahil Khanna.

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
