## [Decision Letter · Decision Letter 0]

19 Oct 2020

PONE-D-20-16731

Avian antibodies (IgY) targeting spike glycoprotein of severe acute respiratory syndrome coronavirus 2 (SARS-CoV-2) inhibit receptor binding and viral replication

PLOS ONE

Dear Dr. Goepp,

Thank you for submitting your manuscript to PLOS ONE. After careful consideration, we feel that it has merit but does not fully meet PLOS ONE’s publication criteria as it currently stands. Therefore, we invite you to submit a revised version of the manuscript that addresses the points raised during the review process.

We look forward to receiving your revised manuscript.

Kind regards,

Paulo Lee Ho, Ph.D.

Academic Editor

PLOS ONE

Journal Requirements:

"The authors have declared that no competing interests exist.   ".

We note that one or more of the authors are employed by a commercial company: 'Scaled Microbiomics, LLC'.

3. Please update your competing interest statement, clarifying whether any commercial competing interests exist in relation to your affiliation with Scaled Microbiomics. For more information regarding PLOS ONE's competing interest policy please visit https://journals.plos.org/plosone/s/competing-interests#loc-what-to-declare

Reviewers' comments:

Reviewer's Responses to Questions

**Comments to the Author**

1. Is the manuscript technically sound, and do the data support the conclusions?

Reviewer #1: Partly

Reviewer #2: Yes

2. Has the statistical analysis been performed appropriately and rigorously? 

Reviewer #1: No

Reviewer #2: Yes

3. Have the authors made all data underlying the findings in their manuscript fully available?

Reviewer #1: No

Reviewer #2: Yes

4. Is the manuscript presented in an intelligible fashion and written in standard English?

Reviewer #1: Yes

Reviewer #2: Yes

5. Review Comments to the Author

Reviewer #1: Reviewer Comments: The following corrections have to be done by the authors for the manuscript approval.

1. Line no.94 – mcg unit has to be corrected as “µg”

2. Line no.101: vaccine preparation – what is the adjuvant used for immunization during sensitization dose and Line no.104: booster immunizations - for booster dose on 14th day of immunizations?

- Clearly mention the antigen-adjuvant ratio preparations for immunization (v/v).

3. Line no.102: each breast of each of one hen pairing – Route of immunization? Intramuscular or subcutaneous has to be mentioned.

4. Line no. 105, 106: Protocols for hen maintenance, immunization, and phlebotomy were approved by the Scaled Microbiomics, LLC Animal Use and Care Committee – Ethical approval reference number need to be mentioned.

5. line no.107: Serum IgY antiviral reactivity:

Serum antibody titre representation using simple bar graphs with clear explanations to show the significant increase in antibody titre levels has to be shown.

6. Line no.124: Total protein concentration of egg yolk and serum:

Standard curve and dynamics of total protein concentration from both egg yolk and serum has to be represented graphically. Even simple graphs are sufficient but it should be clearly self explanatory as per the results obtained.

7. Line no.195: Avian IgY was successfully isolated from yolks, yielding up to 16.8l mg/ml

At which titre day egg this antibody yield was calculated?

8. Line no.196: from serum, yielding up to 23.1mg/ml

Similarly at which titre day of the serum collected post immunization, this antibody yield was calculated?

9. And from both the Line no.195, 196; are antigen-specific IgY antibodies titre has raised as per represented within a couple of week and only with one booster immunization? - JUSTIFY

10. Line no. 177: Plaque reduction assay – Show the results of cell lines and viral plaque titration evaluations.

11. Line no. 211: IgY titers in serum, with titers of 1:10,000 at weeks 3 and 4 (data not shown). – The data has to be represented in graphs showing the dynamics of significant increase in the antibody titration both in serum and the yolk.

12. Line no.166: Binding Inhibition of S1 to ACE2 and Line no. 216: Inhibition of S1 spike binding to ACE2 protein by S1-specific IgY – Show the resulting proof for inhibition. And Western blotting results are must to prove the binding efficacy and specificity of antibodies but it has not been submitted here. Show western blotting results also.

13. Figure 2: Titre of antibodies increased in serum and yolk at the same time period.? How is that possible? Because, it takes a week time for transfer of antibodies from serum to yolk. – Justify with proper results.

Reviewer #2: This manuscript presents excellent evidence for moving forward with an avian antibody approach to the treatment and/or prevention of COVID19. The authors demonstrated a robust and specific IG-Y immune response against SARS-CoV2 S1 spike protein. The approach described in the manuscript is well-conceived and the evidence is compelling. The data presented therefore support the author's conclusions. The statistical analysis is a standard one and it is appropriate for this experimental approach. The underlying PAGE gel data for the statistical analysis is a part of the manuscript.

I do think that discussion should be tweaked a bit: this evidence for this approach as compelling for the treatment of COVID19 based on the use of polyclonal antibodies in other clinical contexts is not germane to a PLOS ONE manuscript. So the discussion should be less speculative.

6. PLOS authors have the option to publish the peer review history of their article (what does this mean?). If published, this will include your full peer review and any attached files.

Reviewer #1: No

Reviewer #2: No

---

## [Author Response · Author response to Decision Letter 0]

2 Mar 2021

Responses to reviewers are contained in the Revised Cover Letter/Rebuttal Letter in tabular form. They are reproduced here but Table formatting is not preserved. Please advise if we should submit this information in a different format.

Original Line Number Reviewer #1 Comments New Line Number Author Response

26-28 (Abstract) Author-initiated change 26-28 Modified language about vaccine availability to reflect recent developments

Introduction, first paragraph Author-initiated change 54-62 Modified language to reflect recent developments on the vaccine front. 

94 mcg unit has to be corrected as “µg” 115 Change has been made

101 vaccine preparation – what is the adjuvant used for immunization during sensitization dose 123 Adjuvant is Montanide ISA 70 VG, specified in new lines 113-114; the phrase “vaccine/adjuvant” has been added for clarity

N/A Clearly mention the antigen-adjuvant ratio preparations for immunization (v/v) 114 This ratio now appears immediately after naming the adjuvant (30% antigen in PBS, 70% adjuvant, v/v). The phrase “30%/70% v/v” has been added to line 125 to reinforce this.

104 booster immunizations - for booster dose on 14th day of immunizations? 126-127 Added this statement: “booster immunizations were performed identically to immunization day 1 for each hen pair.” The awkward word “pairing” was replaced with “pair.”

102 each breast of each of one hen pairing – Route of immunization? Intramuscular or subcutaneous has to be mentioned 124 The word “intramuscularly” has been inserted.

105, 106 Protocols for hen maintenance, immunization, and phlebotomy were approved by the Scaled Microbiomics, LLC Animal Use and Care Committee – Ethical approval reference number need to be mentioned 129 This number now appears in the text. 

107 Serum IgY antiviral reactivity:

Serum antibody titre representation using simple bar graphs with clear explanations to show the significant increase in antibody titre levels has to be shown. 130-139 Please See Notes Regarding Serum IgY Titers Below at ***

124 Total protein concentration of egg yolk and serum:

Standard curve and dynamics of total protein concentration from both egg yolk and serum has to be represented graphically. Even simple graphs are sufficient, but it should be clearly self-explanatory as per the results obtained. 168 The standard curve for total protein concentration in yolk-derived IgY is now provided as new S1 Fig., and is so indicated at Line 168. As noted above, serum collection was discontinued after samples were obtained at Weeks 3 and 4, hence, no intelligible graph of these results is possible. We stress again that this study was intended to determine parameters of yolk-derived IgY, and we do not believe that this missing data impairs that original intent.

195 Avian IgY was successfully isolated from yolks, yielding up to 16.8l mg/ml

At which titre day egg this antibody yield was calculated? 246 ff Four weeks (28 days) after initial immunization, now noted in text at line 244.

196 from serum, yielding up to 23.1mg/ml

Similarly at which titre day of the serum collected post immunization, this antibody yield was calculated? 248 Three weeks (21 days) following the initial immunization, now noted in text at line 246.

195, 196 are antigen-specific IgY antibodies titre has raised as per represented within a couple of week and only with one booster immunization? – JUSTIFY 259 ff The section cited by the reviewer here pertains to antibody concentration, not titer. Under the assumption that the reviewer intended reference to [new] lines 259 ff, where titers are reported, we note that this titer rise is consistent with those seen in our other work on IgY raised against both viral and bacterial antigens; language to this effect is now at Lines 261-264. We routinely provide a single primary and one booster injection spaced 14 days apart. Only if Day 14 titers remain below 1:210 (1:2024) do we re-boost, which was not done in this case given the high initial titers, short duration of study, and high cost of antigen. We provide a graph in a new Supplement (S6 Fig.) at Line 262, similar to Figure 1, showing the rapid rise of egg-derived IgY titers against two other antigens, one bacterial and one viral, to illustrate that the results reported here are typical for our work. In all cases, the IgY titer reported is defined as the maximal dilution returning a signal > 2.1x background. This definition is clarified in Line 204, which now includes the inequality > 2.1 times control.

177 Plaque reduction assay – Show the results of cell lines and viral plaque titration evaluations. 222-241 The reviewer is asking to show the results of the plaque assay in the cell lines. We are providing the neutralizing ab titer dataset in a new Supplementary File (S5 File), indicated in the MS. at Line 242.

211 IgY titers in serum, with titers of 1:10,000 at weeks 3 and 4 (data not shown). – The data has to be represented in graphs showing the dynamics of significant increase in the antibody titration both in serum and the yolk. 260-263 As noted, serum collection was done only at weeks 3 and 4, at both of which time points yolk anti-S1 IgY titers were detectable. Also as noted, this study focused on characteristics of yolk-derived, not serum, IgY.

166; 216 Show the resulting proof for inhibition…. And Western blotting results are must to prove the binding efficacy and specificity of antibodies, but it has not been submitted here. Show western blotting results also. 213-215; 222; 182-183 At 214 ff, we have added information about the S1/ACE2 inhibition kit from Acro Biosystems, and the Method Verification dataset generated prior to the test. The kit is internally validated by the manufacturer; the manufacturer’s Protocol for use in this test is at the URL provided in reference #18, with attention directed to page 9, “Method Verification,” which shows binding of biotinylated human ACE2 protein to immobilized SARS-CoV-2 S protein receptor binding domain, along with a curve showing inhibition of human ACE2/SARS-CoV-2 S protein binding by a standardized inhibitor (anti-SARS-CoV-2 IgG antibody). We believe that this information, along with our existing Figures 3 and 4, address the request for “proof for inhibition.”

Regarding Western Blotting: We used a purified Spike S1 glycoprotein (The Native Antigen Company), rather than an intact viral particle, as the immunogen, for reasons related to biosafety. We therefore believe that Western Blotting would add little useful information beyond the ELISA results in terms of proving the binding efficacy and specificity of antibodies. We have provided a Dot Blot image as a Supplementary Figure (S3 Fig.; line 181/2), which demonstrates that our anti-S1 IgY binds specifically to the purified S1 antigen, with no non-specific binding. We believe that this should be sufficiently confirmatory that the antibodies are specific for their intended targets.

Figure 2 Titre of antibodies increased in serum and yolk at the same time period.? How is that possible? Because it takes a week time for transfer of antibodies from serum to yolk. – Justify with proper results. Figure 2 As noted, serum levels were evaluated only at Weeks 3 and 4 after immunization, by which time specific reactivity was apparent in the yolk-derived IgY samples. We believe this is consistent with the reviewer’s observation, and further, that this study’s focus on characteristics of egg-derived IgY as the therapeutic/prophylactic entity make serum levels less relevant.

 Reviewer #2 Comments 

Discussion I do think that discussion should be tweaked a bit: this evidence for this approach as compelling for the treatment of COVID19 based on the use of polyclonal antibodies in other clinical contexts is not germane to a PLOS ONE manuscript. So the discussion should be less speculative. Text at original lines 284-290, 302-311, and 316-320 has been deleted We believe the reviewer’s concerns center around three paragraphs in Discussion: 284-290, regarding prior use of polyclonal antibodies, have been deleted; 302-311, in which we speculated about economic and other factors that may have prevented uptake of IgY technology, and 316-320, a discussion of use of human polyclonal IgA in intranasal use against upper respiratory infections, have all been deleted. Minor edits to remaining text have been made to assure continuity: deleted “also” in new line 284, Deleted “previous positive” at beginning of new line 295 and inserted “These,” deleted from that line “on intranasal prophylaxis of influenza.”

Acknowledgements From Authors: We inadvertently neglected to acknowledge Dr. Mohamed Ait Ichou, who directs our laboratory. Added Dr. Ait Ichou’s name in Acknowledgements

---

## [Decision Letter · Decision Letter 1]

17 May 2021

Avian antibodies (IgY) targeting spike glycoprotein of severe acute respiratory syndrome coronavirus 2 (SARS-CoV-2) inhibit receptor binding and viral replication

PONE-D-20-16731R1

Dear Dr. Goepp,

We’re pleased to inform you that your manuscript has been judged scientifically suitable for publication and will be formally accepted for publication once it meets all outstanding technical requirements.

Kind regards,

Paulo Lee Ho, Ph.D.

Academic Editor

PLOS ONE

Additional Editor Comments (optional):

Reviewers' comments:

Reviewer's Responses to Questions

**Comments to the Author**

1. If the authors have adequately addressed your comments raised in a previous round of review and you feel that this manuscript is now acceptable for publication, you may indicate that here to bypass the “Comments to the Author” section, enter your conflict of interest statement in the “Confidential to Editor” section, and submit your "Accept" recommendation.

Reviewer #2: All comments have been addressed

Reviewer #3: All comments have been addressed

2. Is the manuscript technically sound, and do the data support the conclusions?

Reviewer #2: Yes

Reviewer #3: Yes

3. Has the statistical analysis been performed appropriately and rigorously? 

Reviewer #2: Yes

Reviewer #3: N/A

4. Have the authors made all data underlying the findings in their manuscript fully available?

Reviewer #2: Yes

Reviewer #3: Yes

5. Is the manuscript presented in an intelligible fashion and written in standard English?

Reviewer #2: Yes

Reviewer #3: Yes

6. Review Comments to the Author

Reviewer #2: The authors have satisfied my concerns and I believe that this manuscript is now ready for publication. The design and execution of the experiments was well conceived. The results are interesting and will help advance our fight against the current pandemic.

Reviewer #3: The authors of this study have evaluated IgY antibodies as potential tool for passive immunotherapy against SARS-COV-2 through blocking the interaction between the virus and ACE2 receptor. There is one concern:

The main point in the manuscript is inhibition of adhesion of viral spike glycoproteins to human ACE2 protein receptors, so, evaluation of binding Inhibition of S1 to ACE2 in methodology section need further details

7. PLOS authors have the option to publish the peer review history of their article (what does this mean?). If published, this will include your full peer review and any attached files.

Reviewer #2: No

Reviewer #3: No

---

## [Editor Report · Acceptance letter]

20 May 2021

PONE-D-20-16731R1 

Avian antibodies (IgY) targeting spike glycoprotein of severe acute respiratory syndrome coronavirus 2 (SARS-CoV-2) inhibit receptor binding and viral replication 

Dear Dr. Goepp:

I'm pleased to inform you that your manuscript has been deemed suitable for publication in PLOS ONE. Congratulations! Your manuscript is now with our production department. 

Kind regards, 

on behalf of

Dr. Paulo Lee Ho 

Academic Editor

PLOS ONE